# Analysis of Precision and Stability of Hand Tracking with Leap Motion Sensor

**DOI:** 10.3390/s20154088

**Published:** 2020-07-22

**Authors:** Aleš Vysocký, Stefan Grushko, Petr Oščádal, Tomáš Kot, Ján Babjak, Rudolf Jánoš, Marek Sukop, Zdenko Bobovský

**Affiliations:** 1Department of Robotics, Faculty of Mechanical Engineering, VSB-Technical University of Ostrava, 70800 Ostrava, Czech Republic; stefan.grushko@vsb.cz (S.G.); petr.oscadal@vsb.cz (P.O.); tomas.kot@vsb.cz (T.K.); jan.babjak@vsb.cz (J.B.); zdenko.bobovsky@vsb.cz (Z.B.); 2Department of Robotics, Faculty of Mechanical Engineering, Technical University of Kosice, 04200 Kosice, Slovakia; rudolf.janos@tuke.sk (R.J.); marek.sukop@tuke.sk (M.S.)

**Keywords:** hand tracking, gesture, leap motion, robot, collaborative robot

## Abstract

In this analysis, we present results from measurements performed to determine the stability of a hand tracking system and the accuracy of the detected palm and finger’s position. Measurements were performed for the evaluation of the sensor for an application in an industrial robot-assisted assembly scenario. Human–robot interaction is a relevant topic in collaborative robotics. Intuitive and straightforward control tools for robot navigation and program flow control are essential for effective utilisation in production scenarios without unnecessary slowdowns caused by the operator. For the hand tracking and gesture-based control, it is necessary to know the sensor’s accuracy. For gesture recognition with a moving target, the sensor must provide stable tracking results. This paper evaluates the sensor’s real-world performance by measuring the localisation deviations of the hand being tracked as it moves in the workspace.

## 1. Introduction

The topic “Human-Robot Cooperation” brings into focus both safety of the robotic workplace (which is critical to keep in mind while designing the workplace), and a way of interaction between the robot and the operator [1,2]. There are several cases where the operator might need to intervene on the robot’s actions, like: [3]:**Safety reasons**—to stop the robot.**Decision making**—choose a robot reaction or a target position.**Trajectory creation**—teaching the robot target positions.**Control of workplace devices**—activating signals, switching valves states.

Standard interaction tools such as touch screens, buttons and other physical control devices do not allow an immersive interaction [4] since they do not provide gesture control, which represents a natural way of human communication. Several contact-less technologies were evaluated based on gesture interaction and localisation [5]. For this experiment, Leap Motion Controller (LMC) was chosen as a device capable of hand detection, tracking and providing information about significant hand elements positions, see Figure 1.

Specifications of the Leap Motion Controller are:**Outer dimensions:**80×30×13 mm.**Data connection:** USB 2.0.**Interaction zone:** up to 60 cm extending from the device in $50^{{}^{\circ}}\times 120^{{}^{\circ}}$ field of view.**Cameras:** Two 640×240 pixel near-infrared cameras spaced 40 millimetres apart which operate in the 850 ± 25 nm spectral range; typically operates at 120 Hz but the hardware is capable of 240 Hz or more.**Hand tracking output:** 27 distinct hand elements, including bones and joints.

The LMC sensor uses two infrared stereo cameras with three infrared LEDs (Light Emitting Diode) illuminating the monitored space; however, the details of the sensor function are patent protected. The depth information is estimated based on a stereoscopic view of the scene. Several technical limitations are present due to the technology of the sensor. The cameras have a 150${}^{{}^{\circ}}$ wide field of view which results in a significantly distorted image. The use of infrared brings its limitations depending on lighting and other environment conditions as well as on cleanliness of the glass cover. Another problem of this sensor is the limited range of the illumination provided by three low-power inbuilt LEDs. The interaction box [6] is a boundary area, where the expected accuracy of the sensor is the highest. The positive *y*-axis represents the depth values in the view direction of the sensor; the minimum distance along the *y*-axis is 82.5 mm, and the maximum distance is 317.5 mm. The *z*-axis of the interaction box, which is perpendicular to the longer side of the sensor, is in the range from −73.5 mm to 73.5 mm and the range along the *x*-axis is from −117.5 mm to 117.5 mm.

The manufacturer offers Leap Motion Orion SDK (Software Development Kit, Ultraleap, Mountain View, CA, US), which provides the LeapC API (Application Programming Interface, Ultraleap, Mountain View, CA, US) for communication with the sensor. It is possible to work with the image stream and distortion parameters or to access the hand tracking data directly which includes simple gesture recognition features along with access to positions of individual finger joints, the palm’s location, its normal and velocity.

This research follows up on the study of intuitive robot control systems [7]. An example scenario in which LMC could be used together with a collaborative robot is depicted in Figure 2a where the robot is programmed to operate a tool to manipulate screws on a given target. In this example the robot is tasked to disassemble the target object for which the position of screws are not predefined. The collaboration process is divided into 3 stages, they are:**Scene identification:** At this stage an RGB-D (Red Green Blue-Depth) camera detects selected entities (e.g., screws) and projects the result to the visual feedback device.**Interaction:** This step involves using the data from LMC. Operator’s hand is tracked and the position of the fingertip is checked against the positions of detected entities. The data can also be used to select highlighted entities, perform gesture based commands, define drop areas (for pick and place applications) or to redefine the area of interest for entity recognition by the RGB-D camera.**Robot operation:** After selecting the tasks to perform, the robot program is compiled from pre-defined operations (screwing, unscrewing) and positions of entities based on RGB-D data, marked by the user’s fingertip.

In the Figure 2b is an example of an augmented image, where detected entities are highlighted in the camera image. To be able to indicate a target for the robot, it is necessary to track the index fingertip in the space. Position of the tip is important for interaction with detected objects and tracking stability for stable feedback in the mixed reality image for intuitive control. Based on the measurement results collected in this article we can specify corrections and tolerance margins of positions detected with LMC.

## 2. State of Art

The Leap Motion Controller has previously been compared with a standard input device used in pointing applications (e.g., mouse) [8]. This research draws attention to the significant error observed when the device is used by a novice user. In addition, a position detection was performed with a high accuracy motion tracking system and compared with LMC in which it was found that LMC is not suitable for professional tracking yet [9]. The researchers in the past have performed experiments to validate the accuracy of the system in both static and dynamic (with moving target) contexts. The measurements were made with the sensor pointed up towards the tracked target. The motionless scene was comprised of a plastic model of an arm. In contrast, the dynamic measurements were based on tracking of two moving spheres. The results showed deviations of the position to be lower than 0.5 mm in the static scenario. When making measurements in the dynamic environment, high accuracy drops were observed when the measurements were made outside the interaction box defined by the LMC. The most stable hand pose for tracking was also determined [10], for which measurements were done with 16 different hands, assuming different orientations with different gestures. The accuracy of hand recognition was evaluated with two types of hand gestures and various orientations of the wrist.

LMC has been analysed in a wide spectrum of applications including virtual grasping [11], where the absolute accuracy of the tracked hand is not of the main concern. The sensor has been evaluated for training laparoscopic surgeries as well [12]. LMC has been used in the past in robotic arm control applications [13,14], where the data from the sensor was processed on a computer and then transmitted to the robot controller. The standard output from the sensor API can be anything from the simple position of the hand or a finger to positions of all the joints of the hand being tracked simultaneously and can be used to control a suite of mechanisms including an anthropomorphic gripper [15] in which case the recognized gesture is processed, and the gripper is actuated accordingly. A single sensor can be used to recognize parts of the hand, which is not being occluded by any other object in the environment. The pose can sometimes be retrieved even when the whole hand is not in the frame based on tracking what is visible, or multiple cameras [16] can be used to improve the acquired information. This study describes a fusion of data from two sensors in an optimal position to cover the workspace without losing information about hand gesture when one sensor does not have enough information to detect the gesture.

## 3. Measurement Setup

The measurements were performed with LMC mounted above the workplace and pointed downwards. This does not represent a standard desktop setup for LMC. However, it is similar to the head-mounted setup when using the sensor with a Virtual Reality (VR) headset. This setup is also closer to the intended application where the sensor is mounted with the overview camera above the workplace.

To detect pointing gestures, a software tool was developed that allowed us to fetch the positions of the palm and the tip of the index finger localized by the hand tracking system.

During the first measurement setup the position data were obtained by tracking the hand following a square trajectory marked by a black tape on the working table with the camera mounted 330 mm above the table, this corresponded to 317.5 mm, which was the maximal recommended detection distance of the sensor. The index finger followed the square trajectory elevated to different distances from the sensor. The best results were obtained at a distance approximately 240–250 mm below the sensor. Three main problems were observed:It was difficult to move hand repeatedly along the same trajectory. Thus the collected data could not be compared, as can be seen in Figure 3.Due to the unsteady frame rate, it was difficult to map the tracked position accurately on a time basis.The LMC was not able to reliably separate the hand from the background when the hand was too close to the table surface.

Greater stability was achieved by raising the hand above the table as can be seen on Figure 4a—this simplified the background subtraction for the device. The data from five consecutive measurements with the same conditions are visualized in Figure 3. However, the deviation of the recognized finger positions during measurements was unacceptable; in the *x* and *z* axes the measured values oscillated in a 2–4 cm wide area around the target value and the depth values had an 8 cm scatter. High oscillations during the movement from the point [0,0.06] to the point [0,−0.06] in the sensor [X,Z] coordinates were caused by unstable hand recognition against the table surface.

To ensure repeatability of the measurement and precise hand movements, a measuring system was implemented using an industrial collaborative manipulation arm Universal Robots UR3 mounted on the table. A 3D-printed hand corresponding to the human hand was mounted to the flange of the robot arm, see Figure 5. This 3D-printed hand is based on a 3D scan of a real human hand with actual dimensions. It was printed using white PLA plastic which in our measurement scenario, shows the same tracking results as the real hand. The printed hand was reinforced with an expansion foam to increase stiffness and avoid unwanted vibrations. The UR3 robot was a six degree of freedom angular robot with a maximum reach of 500 mm. This range limitation led to the decision of conducting the measurements only along the positive *x*-axis of the sensor with the assumption of axis symmetry.

The Leap Motion Controller was placed above the table surface pointed down, as can be seen on Figure 4b. The LMC was elevated from 330 mm to 530 mm to ensure reaching all the positions of the measured workspace with the robot—Figure 4c. The orientation of the controller relative to the table was adjusted using the Intel RealSense D435i camera mounted next to the sensor (Figure 4b). Using the depth quality tool, the sensor was positioned perpendicular to the table surface and reinforced with a stiff frame. The surface of the table was covered with a non-glossy cover to minimize the reflected ambient light. The orientation of the sensor was according to the recommended LMC setup, meaning the positive *z*-axis was pointing towards the side with the robot. The rotation of the sensor along the vertical axis was set according to the robot base coordinate system, so that the positive *z*-axis of the sensor corresponds to the positive *y*-axis of the robot. The position and orientation in the case of an ideal calibration are described with the following transformation matrix:(1)$$T_{R_{base}}^{C}=\left[\begin{array}{cccc} 1 & 0 & 0 & -0.22\\ 0 & 0 & 1 & -0.27\\ 0 & -1 & 0 & 0.53\\ 0 & 0 & 0 & 1 \end{array}\right]$$

By extending the transformation matrix $T_{R_{base}}^{R_{TCP}}$ from the robot base to the robot TCP (Tool Centre Point) by the position of the index fingertip [0.02,0.06,0.37] in the TCP coordinate system, we get the transformation matrix $T_{R_{base}}^{I_{tip}}$ from the robot base to the fingertip:


(2)$$T_{R_{base}}^{I_{tip}}=T_{R_{base}}^{R_{TCP}}.\left[\begin{array}{cccc} 1 & 0 & 0 & 0.02\\ 0 & 1 & 0 & 0.06\\ 0 & 0 & 1 & 0.37\\ 0 & 0 & 0 & 1 \end{array}\right]$$


The measurement was performed by moving the printed hand mounted to the robot at a specific distance from the sensor. To cover the majority of the view range of the sensor, the best intersection of the robot working area and the sensor interaction box was found. The robot performed linear movements parallel to the coordinate axes of the sensor. The measured positions of the robot flange were defined as reference positions of the hand in the robot coordinate system, as shown in Figure 6. The robot positions were represented by values logged from the robot during the measurements. Positions of the endpoint are calculated based on the joint angles using forward kinematics. The values in the Figure 6 are shown in the robot base coordinate system.

Table 1 shows calculated standard deviations from the target value of all traverses, and the maximum value is in the table. The sensor depth value (the *z*-axis of the robot) deviation is calculated from the whole trajectory.

The measurement was carried out in the standard room conditions with 20 ${}^{{}^{\circ}}$C and artificial lighting to eliminate direct sunlight which could affect the sensor performance. These conditions followed the sensor requirements since the goal of the measurement was to obtain the performance of the sensor under standard conditions. The expected use of this application is in a workplace with a collaborative robot assisting the operator during an assembly task. This scenario did not consider dust, smoke, dirt, fog or other disturbing conditions which could negatively affect the sensor.

This application was not meant as a safety layer of the workplace, meaning there was no requirement for a robust real-time hand tracking. The analysis of the behaviour with different hand movement speeds was done to determine whether the application can track the hand under various speeds. A stable hand tracking was necessary for the position feedback and thus, a user-friendly operation.

## 4. Position Accuracy of the Sensor

The evaluation of the recognition accuracy of the LMC was conducted based on the result of an experiment where the robot moved the 3D-printed hand across the workspace. The movement trajectory consisted of linear segments. Two different measurements were performed—each at a specific distance from the sensor. To measure the accuracy along the *x*-axis, the robot moved the hand in 250 mm long traverses with 30 mm spacing across the captured area. Similarly, for the *z*-axis, the robot performed 200 mm long traverses with 30 mm spacing. The length of the movement and spacing was selected within the working range of the robot. The motion parameters of the robot were chosen to avoid vibrations during the movement:**Velocity of the TCP:** 10 mm/s**Acceleration of the TCP:** 5 mm/s^2^

An application based on LeapC library logged data from the sensor and evaluated the position of the tip of the index finger to recognize the pointing gesture.

Deviations from the expected positions gathered during the first measurement (250 mm below the camera) are presented in Figure 7. This graph represents evaluated data from both movements describing the [X,Z] position and corresponding deviations for every movement in the side graphs. The green zone represents the interaction box. Individual traverses of both movements are numbered in the figure corresponding to Table 2.

The box plots represent a statistical evaluation of every traverse. They are aligned to the corresponding traverses in the 2D position graph. The zero value of the side box plots represents the mean value of the traverse. Table 2 shows standard deviations of individual traverses in the *x* and *z*-direction and the corresponding depth (*y*-axis) deviations. 2D position graph depicts nine traverses in the direction perpendicular to the *x*-axis numbered from 1 (the closest to the *z*-axis) to 9 (the farthest in the positive *x*-direction). The graph also depicts 11 traverses perpendicular to the *z*-axis, where number 1 is the furthest in the positive *z*-direction, and number 11 is the furthest in the negative *z*-direction.

Based on the deviations provided in Table 2, it is evident that the *X* values within the interaction box (traverses 2 and 3 in Table 2) were the closest to the target positions. The *Z* values had an irregularity in the fifth traverse; however, as expected, the values in traverses 4 to 8 were stable. The most accurate results were in the negative *z*-axis which might have been caused by higher tracking stability when the whole hand (including the forearm) was within the sensor’s FOV.

Figure 8a,b represents deviations from absolute target positions, confirming the highest accuracy within the interaction box. The deviations of the *Z* values were not symmetric due to better tracking results with a higher percentage of the hand surface located within the sensor’s field of view.

The absolute value of depth (*y*-axis) and deviation from the mean value (*y* deviation) is displayed for *Z* and *X* movements in Figure 9a,b. The individual traverses are bounded with the coloured column, and the colour corresponds to the position data in Figure 7. Due to the fact that the direction of sequential measurements is opposite to the interpretation style in tables, the measurements are ordered decreasingly. During the traverses, data were logged approximately each 8 ms.

The measurements show that the deviations of the depth values derived from stereo-vision estimation were significantly larger than the deviations of values in the *X*-*Z* plane.

The second level of measurement was defined to 200 mm below the sensor. The measurement process was the same as in the first case, except for the first traverse along the *z*-axis was skipped because of the range limitation. Due to the same reason, the last two traverses in the *x*-direction were shortened by 30 mm. Figure 10 represents the results and values of standard deviations of individual traverses in Table 3. The most distant *X* traverses from the *z*-axis provided incomplete data due to the loss of tracking.

By comparing the values in the two examined levels, we can see that the data were less stable at the height closer to the sensor, especially outside the interaction box. Within the interaction box, the data were comparable at both heights.

The depth values are shown in Figure 11a, where the data from traverses are illustrated along the *x*-axis, and Figure 11b with *z*-direction traverse values. Depth deviations within the interaction box were slightly lower than at the 250 mm level, on the other hand there was a significant deterioration in the outer area. Standard deviations of [X,Z] positions for the *X* traverses in both measurements within the interaction box were lower than 1.5 mm and for the *Z* traverses lower than 3 mm. The higher value results from covering more distant areas from the interaction box within the traverse. In the outer area, the value did not exceed 4 mm in the 250 mm level and 8 mm in the 200 mm level. Standard deviations of depth values are within 6 mm for all measurements, showing no direct relation with the position inside or outside the interaction box.

## 5. Tracking Stability

Tracking stability was evaluated for a range of motion parameters of the robot, as shown in Table 4. The measurement procedure was repeated five times for each set of linear velocity and acceleration. We utilized the same movement pattern as the one used during the measurement of the position accuracy.

To compare the stability of tracking data, 3 traverses along the *x*-axis with different *Z* values were chosen to compare deviations with a different velocity of the tracked object.

**Second traverse** in positive *z*-space—Figure 12a**Fifth traverse**, which is the closest to the *x*-axis—Figure 12b**Tenth traverse**—Figure 12c in the negative *z*-space.

Table 5 shows the comparison of standard deviations of *Z* values of three chosen traverses for the five different robot motion settings, as defined in Table 4. Deviations in individual settings had no significant differences, meaning the frame rate of the sensor is sufficient for covering the whole range of hand movement speed.

The deviations were lower and manifested a similar magnitude when the hand was within the interaction box of the sensor. As in the previous measurement, lower deviations appeaedr more frequently in the negative *z*-space than in the positive *z*-space due to a higher percentage of the hand surface presented within the sensor range.

## 6. Conclusions

Leap Motion Controller was primarily developed to be an intuitive input device for computers and VR headsets. In this paper, we attempted to generalize the usage of the sensor in robot control applications and we performed a benchmark of the camera capabilities through different test cases. The device performs well when the measurements are made at a level of 250 mm below the sensor, when the tracked hand is still in the interaction box, the deviation of the detected position is around 5 mm which is sufficient to track the pointing gesture. Although, outside this box, the error can be up to 10 mm and the same is true when measurements are made at a distance of 200 mm from the sensor. These measured values are acceptable for the target purpose, and through the experiments, the conclusion was reached that the sensor can provide stable measurements even when operating outside the recommended measurement range.

It was also observed that the high frame rate of tracking makes the gesture recognition being sufficiently reliable when the arm is moving at different speeds in the measurement window.

In subsequent work, we plan to use the gathered data from the sensor to develop correction models to improve the output directly by characterising the deviations.

In this work, the measurements were carried out under optimal conditions relating to the posture of the hand being tracked—an open hand gesture with no pitch or yaw rotation was used. In future research, we plan to perform these measurements on a variety of different hand models varying according to their size and pose to get even closer to real-world situations.

## Figures and Tables

**Figure 1 sensors-20-04088-f001:**
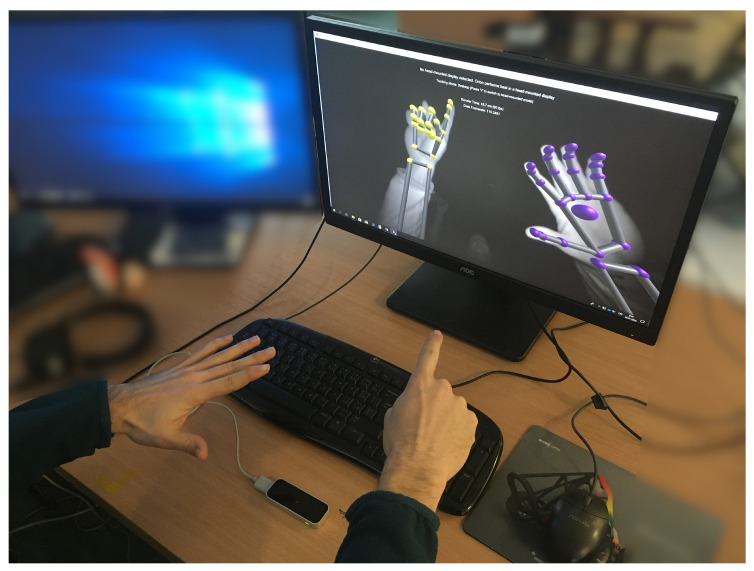
Leap Motion hand detection using the Leap Motion Visualizer.

**Figure 2 sensors-20-04088-f002:**
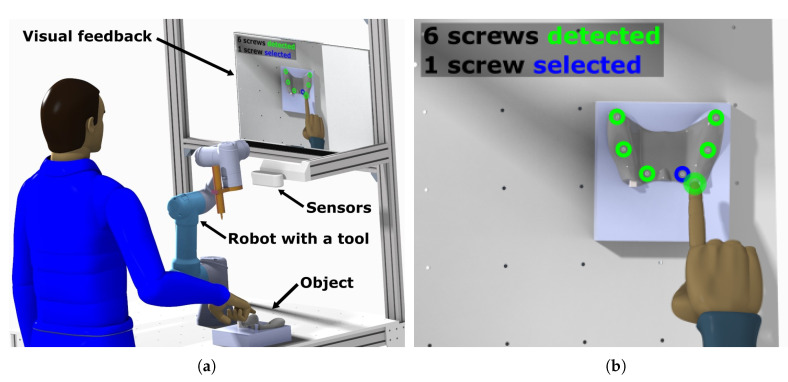
Example scenario of using the Leap Motion Controller (LMC) with the robot: (**a**) Workplace with LMC, overview camera and a collaborative robot, (**b**) visual feedback—an augmented reality image with an operator’s index fingertip highlighted based on LMC data.

**Figure 3 sensors-20-04088-f003:**
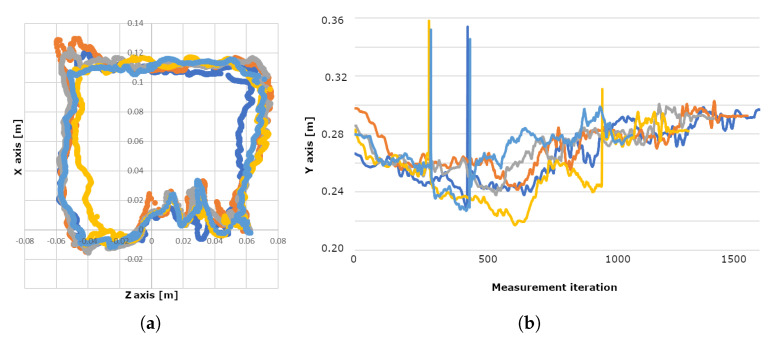
Visualisation of the trajectory of index fingertip while following the square path—five colours corresponding to five measurements: (**a**) Position of the fingertip in X–Z (horizontal) plane, (**b**) Y coordinate during the measurement.

**Figure 4 sensors-20-04088-f004:**
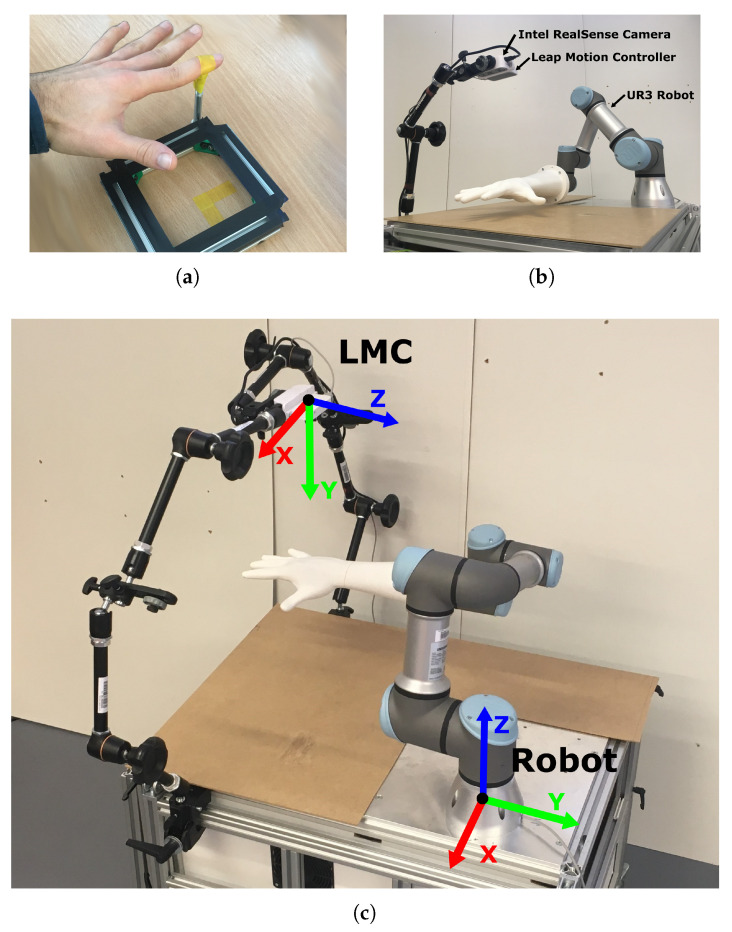
Evolution of finger position measurement setup: (**a**) Basic setup with finger following square template, (**b**) Setup with a camera mounted above the robot workspace, (**c**) Robust camera mounting after calibration with LMC coordinate system.

**Figure 5 sensors-20-04088-f005:**
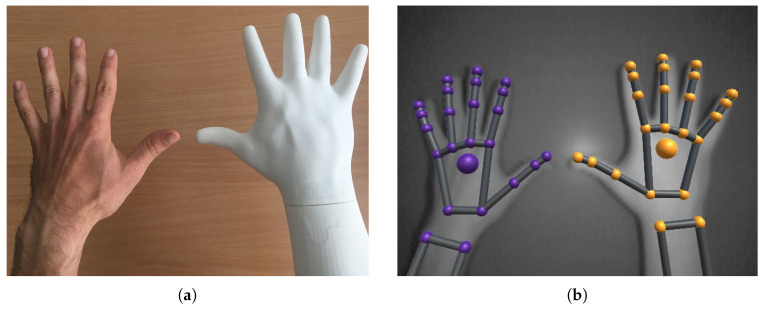
3D-printed hand corresponding to the human hand. (**a**) Colour image of a human and plastic hand, (**b**) Image from the sensor with both hands detected.

**Figure 6 sensors-20-04088-f006:**
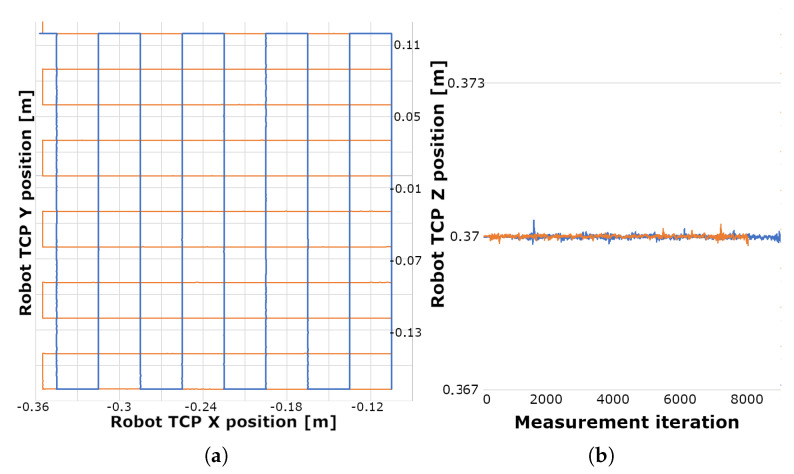
Robot positions logged during the movements (reference values); the blue line represents the movement with constant *X* coordinate and the orange line with constant *Y* coordinate in the robot coordinate system: (**a**) robot position in the *X*-*Y* (horizontal) plane, (**b**) the *Z* coordinate during the measurement.

**Figure 7 sensors-20-04088-f007:**
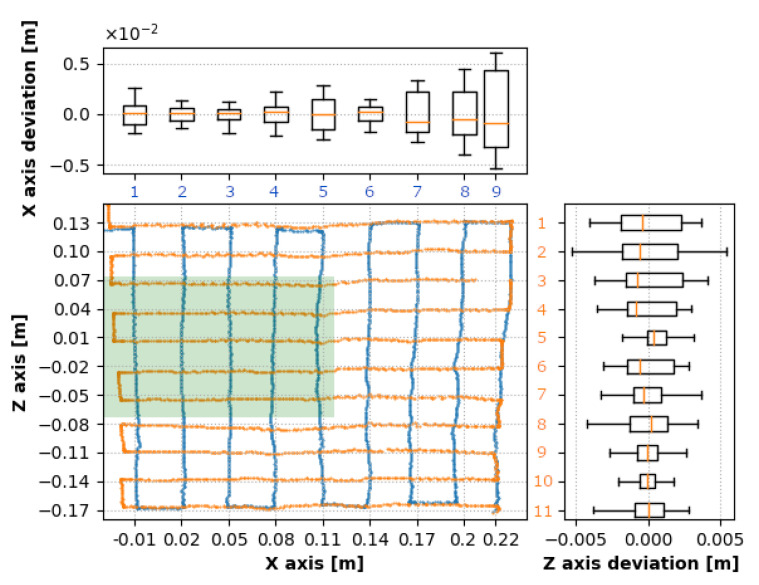
Deviation in the level 250 mm: orange dots represent results during traverses with a constant value in the *z* coordinate of the fingertip, the blue dots represent results during traverses with a constant *x* coordinate of the fingertip. The side box plots correspond to individual traverses and present the overall deviations during each individual traverse.

**Figure 8 sensors-20-04088-f008:**
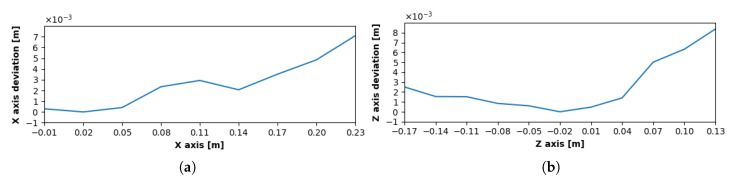
Deviation from the absolute target position in the level 250 mm: (**a**) Movement in the direction perpendicular to the *x*-axis, (**b**) Movement in the direction perpendicular to the *z*-axis.

**Figure 9 sensors-20-04088-f009:**
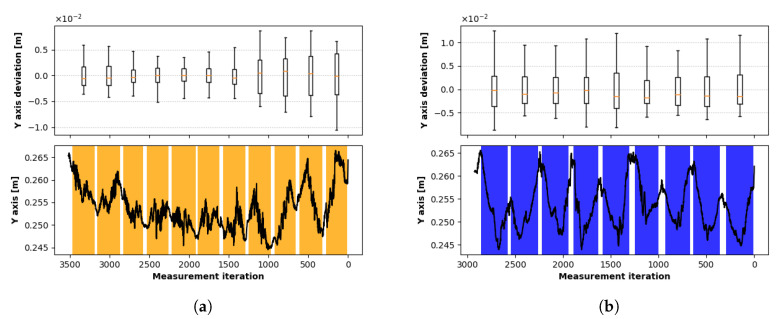
Deviation in Y (vertical) axis during traverse at the level 250 mm: (**a**) traverse in the direction perpendicular to Z axis, (**b**) Traverse in the direction perpendicular to X axis.

**Figure 10 sensors-20-04088-f010:**
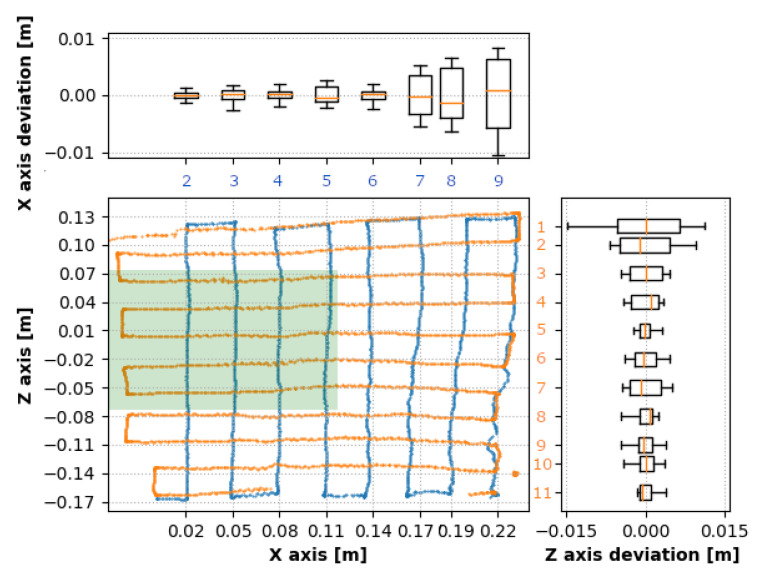
Deviation in the level 200 mm: the orange dots represent results during traverses with a constant value in the *Z* coordinate of the fingertip, the blue dots represent results during traverses with a constant value in the *X* coordinate of the fingertip. The side graphs correspond to individual traverses and present overall deviations during the traverse.

**Figure 11 sensors-20-04088-f011:**
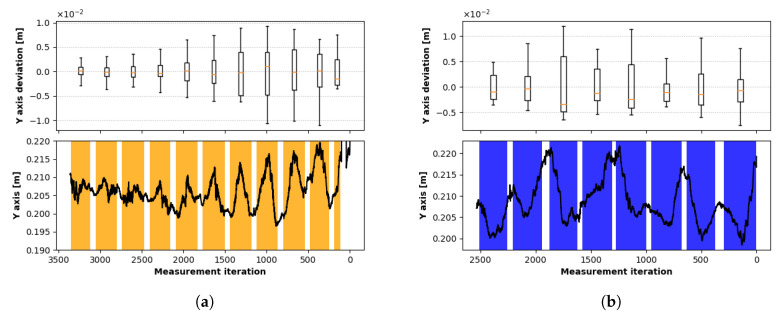
Deviation in the *y* (vertical) axis during movement in the level 200 mm: (**a**) Movement in the direction perpendicular to the *z*-axis, (**b**) Movement in the direction perpendicular to the *x*-axis.

**Figure 12 sensors-20-04088-f012:**
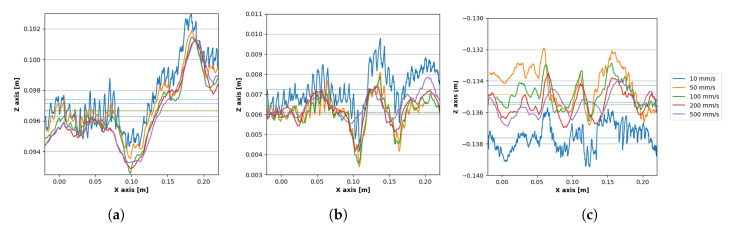
Positions of the fingertip during movement with a different linear velocity of the hand in the level 250 mm: (**a**) second traverse, (**b**) fifth traverse, (**c**) tenth traverse.

**Table 1 sensors-20-04088-t001:** Maximum standard deviations of the robot position.

Traverse	X [mm]	Y [mm]	Z [mm]
X traverse	$4.4\times 10^{-2}$	-	$2.6\times 10^{-2}$
Y traverse	-	$3.7\times 10^{-2}$	$2.9\times 10^{-2}$

**Table 2 sensors-20-04088-t002:** Standard deviations of each traverse in the level of 250 mm.

	X [mm]	Y_X_ [mm]	Z [mm]	Y_Z_ [mm]
1	1.1	5.6	2.2	2.5
2	0.68	3.9	2.3	2.5
3	0.63	4.4	2.1	1.8
4	0.94	4.2	1.8	1.8
5	1.5	4.8	2.7	1.6
6	0.87	4.2	1.7	1.8
7	1.9	3.9	1.4	2.3
8	2.3	4.4	1.7	3.8
9	3.7	4.0	1.2	4.2
10	-	-	0.8	4.6
11	-	-	1.2	4.6

**Table 3 sensors-20-04088-t003:** Standard deviations of each traverse in the level of 200 mm.

	X [mm]	$Y_{x}$ [mm]	Z [mm]	$Y_{z}$ [mm]
1	-	-	7.3	1.5
2	0.6	2.6	5.2	1.4
3	0.99	3.3	3.0	1.4
4	0.92	5.9	2.6	2.1
5	1.4	3.7	1.3	2.8
6	1.1	5.2	2.5	3.3
7	3.5	3.6	2.9	4.5
8	4.3	4.4	1.8	5.6
9	6.1	4.9	1.8	5.3
10	-	-	2.0	4.3
11	-	-	1.5	3.6

**Table 4 sensors-20-04088-t004:** Robot linear movement limits.

Measurement	TCP Velocity [mm/s]	TCP Acceleration [mm/s^2^]
1	10	5
2	50	500
3	100	1000
4	200	1000
5	500	1000

**Table 5 sensors-20-04088-t005:** Standard deviations of chosen movements in the level of 250 mm with different velocities.

Measurement	Traverse 2 [mm]	Traverse 5 [mm]	Traverse 10 [mm]
1	2.5	0.9	0.8
2	2.0	0.8	1.2
3	2.1	0.8	0.8
4	2.2	0.8	0.8
5	2.3	0.7	0.7

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
