# Peer review of "Analysis of Precision and Stability of Hand Tracking with Leap Motion Sensor"

_sensors, 2020, doi:10.3390/s20154088_

Round 1

Reviewer 1 Report

I like the experiments with the printed hand. However, the paper gives no convincing example of a scenario where the accuracy of the automatic interpretation of the pointing gesture, as well as of the hand trajectory are important. Please add such an example, its justification, and results of experiments with related hand poses and trajectories.

Author Response

The authors appreciate all the reviewers’ and editors’ comments. Thank you for the careful reading of our manuscript and your valuable suggestions. The comments are fair, encouraging and constructive. After considering the comments carefully, we have revised the manuscript accordingly. All changes (including the changes added in reaction to the other reviewers) are highlighted in yellow in the article. The whole text went through the in-depth English check and several parts were rewritten for the better readability.

We added a paragraph starting at the line 52 which describes a purpose of the measurement. The goal is to find a sensor or a technology for intuitive interaction with the robot. Added paragraph is supported with an illustrative image and with a reference to the previous research. Results corresponding to the implementation of the sensor to the proposed application are not ready yet.

Methods and results were supported with better descriptions of the coordinate systems (Figure 4), specification of measurement conditions starting at line 170. All the text was revised for better understandability and readability.

Reviewer 2 Report

The hand tracking is a relevant and very intuitive way to control tool purposes. In this sense, the assessment of precision and stability proposed by authors is necessary for applications on human-robot interaction. The work presents the static and dynamic evaluation of hand tracking by the LMC system for two specific levels in relation to the distance from the sensor, for both a user's hand and a robotic hand. In order to improve the manuscript, I leave for authors to consider the following suggestions.

The introduction is mostly reduced to the specifications of the LMC device. However, a wide context of the work and its main contribution are suggested to be included. 

I understood the purpose of demonstrating that the experiment performed by a robot would lead to more objective evaluation. Moreover, it is implicit that consecutive trials in the experiment following square trajectory were performed by only one subject. Because the purpose of the application involves human hand, it would be expected more trials from different volunteers.

I noted that two different setups for the camera mounted above the table were used for human and robotic hand (330 and 530 mm respectively). I suggest explaining this choice.

Figures with deviations of trajectories both for XY and Z axis, as well as box graphs, were presented on different scales, which makes difficult to compare experiments on both levels. Moreover, a comparison of the two levels is hurriedly described in lines 182-183. A wide discussion and statistical comparisons would be desired.  

Minors concerns

-Please, consider an expert revision of English.

-In line 162, the sentence is unclear. What do you mean by (2,3)?

-In line 172, the corner [0.22,-0.17] defined as the start point should be indicated before in text (~152) for easier understanding.

-In section 4 was indicated that two measurements will be performed. It may be convenient to indicate in the text that the presentation of the first measurement results begins in line 152.

-Figure 4 is cited in the text before citing Figure 3. Should be renumbered

-Define the abbreviation TCP.

-Please, use dot (.) as a decimal separator

Author Response

The authors appreciate all the reviewers’ and editors’ comments. Thank you for the careful reading of our manuscript and your valuable suggestions. The comments are fair, encouraging and constructive. After considering the comments carefully, we have revised the manuscript accordingly. All changes (including the changes added in reaction to the other reviewers) are highlighted in yellow in the article. The whole text went through the in-depth English check and several parts were rewritten for the better readability.

We added a paragraph starting at the line 52 which describes a purpose of the measurement. The goal is to find a sensor or a technology for intuitive interaction with the robot. Added paragraph is supported with an illustrative image and with a reference to the previous research. Results corresponding to the implementation of the sensor to the proposed application are not ready yet.

The measurement with a human hand provided incomparable detection results. It was hard to ensure reference position for deviation calculation. That is why real hand was replaced with an 3D printed one which can be precisely positioned. In the first stage of the experiment this scenario was exchanged with the robot-based measurement. This is a reason why the research was not extended to other participants.

Different mounting setups of the camera were explained in the text (line 96, line 123). Distances correspond to optimal sensing range of the sensor and the elevated height 530 mm corresponds to optimal intersection of sensor range and robot reach areas.

The scale of the box graph is different for better readability of the results. Unification of the scale would cause flat box graphs with unclear values. Statistical data is described more closely on the line 210.

All minor concerns were revised and fixed:

  • (2,3) was explained on the line 183 with the reference to the table
  • Traverse counting was implemented to the figure which was specified on the line 173
  • On the line 170 was specified the measurement level which indicates the first measurement
  • Figures 4 and 3 in original text (5 and 4) in new text were exchanged and the numbering in the text was corrected
  • Abbreviation TCP was defined on the line 133
  • All decimal separators were exchanged for dots including figures 3 and 6

Reviewer 3 Report

Evaluating sensor accuracy and stability is a very interesting yet hard problem. Particularly it is hard to gather ground truth of hand poses and positions. I am very excited to see that there are researchers working on this.

This paper focuses on Leap Motion Control (LMC) sensors. To accurately control the hand position, the authors actually printed a 3D hand model to use. Mean and standard deviation errors are reported during the tests. 

Here are my comments:

  1. This is a very interesting paper, but it is not easy to read. There are plenty of English grammar issues that bothers me from understanding.
  2. X-axis, Y-axis and Z-axis are frequently used in paper. But the authors didn't describe the coordinate setup. Is it global coordinate, or local to the LMC. Is it a z-up coordinate system or y-up? It would be clearer if the authors can just illustrate it in Fig 4. 
  3. I am skeptical about using 3D printed hand model vs. using a real human hand. The printed hand follows one real hand, thus its dimension is fixed. It is hard to cover all hand sizes in the real world. Particularly finger length is a key factor for accuracy of pointing gestures. The author also reported using the 3D printed hand in white color. Thus I am curious if the results are the same using hand with regular skin tone.
  4. From the reported results, what can be learned? How to apply the findings to practical applications. It would be better for authors to summarize key takeaways.

In general, I think this paper presents a very interesting topic. Maybe it just need one extra iteration to polish the writing.  

Author Response

The authors appreciate all the reviewers’ and editors’ comments. Thank you for the careful reading of our manuscript and your valuable suggestions. The comments are fair, encouraging and constructive. After considering the comments carefully, we have revised the manuscript accordingly. All changes (including the changes added in reaction to the other reviewers) are highlighted in yellow in the article. The whole text went through the in-depth English check and several parts were rewritten for the better readability.

  1. Coordinate systems were added to the figure 4c and it was checked in the text that referencing to the axes is clear
  2. LMC is not based on colour image but on near infrared stereo image. According to the sensor specification, different hand shape might cause problem, but colour should not. This article is not about testing the LMC working conditions. Plastic hand worked during testing with the same stability as the real hand. Figure 5 was extended with comparison of the real hand and artificial hand detection.
  3. We added a paragraph starting at the line 52 which describes a purpose of the measurement. The goal is to find a sensor or a technology for intuitive interaction with the robot. Added paragraph is supported with an illustrative image and with a reference to the previous research. Results corresponding to the implementation of the sensor to the proposed application are not ready yet. We found out that this sensor provides sufficient data for robot interaction applications.

Round 2

Reviewer 1 Report

In my first review I wrote: “The paper gives no convincing example of a scenario where the accuracy of the automatic interpretation of the pointing gesture, as well as of the hand trajectory are important. Please add such an example, its justification, and results of experiments with related hand poses and trajectories.”

I have not found a satisfactory answer to this problem, which I consider important. Please describe the scenario, the protocol for communicating with the robot using gestures and the required accuracy of interpretation of these gestures.

Author Response

Dear Reviewer,

thank you for your suggestion. We added a description of the example scenario starting on the line 52. It is supported with new figures 2a and 2b which substitute the figure 2. This study gives us data for hand tracking accuracy evaluation and hand tracking stability during movements of the hand. This is important for the second stage of the proposed application, where we need to localize the fingertip as well as to provide the feedback to the user for an intuitive interaction.

Reviewer 2 Report

Thank you for accepting the suggestions. All pointed out suggestions were considered and solved, and I suggest the publication of the manuscript.

Author Response

Thank you